# Towards Exploitation of Adaptive Traits for Climate-Resilient Smart Pulses

**DOI:** 10.3390/ijms20122971

**Published:** 2019-06-18

**Authors:** Jitendra Kumar, Arbind K. Choudhary, Debjyoti Sen Gupta, Shiv Kumar

**Affiliations:** 1Indian Institute of Pulses Research, Kalyanpur, Kanpur 208 024, Uttar Pradesh, India; debgpb@gmail.com; 2ICAR Research Complex for Eastern Region, Patna 800 014, Bihar, India; 3Biodiversity and Integrated Gene Management Program, International Centre for Agricultural Research in the Dry Areas (ICARDA), P.O. Box 6299, Rabat-Institute, Rabat, Morocco

**Keywords:** adaptive traits, gene/QTL, epigenetics, transgenics, genome editing, climate-smart pulses

## Abstract

Pulses are the main source of protein and minerals in the vegetarian diet. These are primarily cultivated on marginal lands with few inputs in several resource-poor countries of the world, including several in South Asia. Their cultivation in resource-scarce conditions exposes them to various abiotic and biotic stresses, leading to significant yield losses. Furthermore, climate change due to global warming has increased their vulnerability to emerging new insect pests and abiotic stresses that can become even more serious in the coming years. The changing climate scenario has made it more challenging to breed and develop climate-resilient smart pulses. Although pulses are climate smart, as they simultaneously adapt to and mitigate the effects of climate change, their narrow genetic diversity has always been a major constraint to their improvement for adaptability. However, existing genetic diversity still provides opportunities to exploit novel attributes for developing climate-resilient cultivars. The mining and exploitation of adaptive traits imparting tolerance/resistance to climate-smart pulses can be accelerated further by using cutting-edge approaches of biotechnology such as transgenics, genome editing, and epigenetics. This review discusses various classical and molecular approaches and strategies to exploit adaptive traits for breeding climate-smart pulses.

## 1. Introduction

Pulses are cultivated worldwide as major or minor crops (Table 1) to provide for the nutrition and livelihood of millions of peoples. Pulses, being a rich source of protein (22–26%) and micronutrients (especially Fe and Zn), are a balanced food for vegetarians when complemented with cereals. Also, the green and dry plant parts of these crops are used as feed and fodder in many livestock production systems [1], and their cultivation has long helped to sustain cereal-based cropping systems through biological nitrogen fixation and carbon sequestration [2]. Most of these pulses originated in the Mediterranean region [3]. The reproductive phase of most such crop plants often occurs in the dry climate of the Mediterranean region during spring. This favors the evolution and survival of plants with “cleistogamous” flowers, as cleistogamy prevents desiccation of anthers and stigmas and encourages full seed set by autogamy [4]. Cleistogamy of pulses appears to be a relic of evolutionary antecedents. However, such cleistogamous flower buds do open for a small period, providing opportunities for occasional natural outcrossing, which occurs in almost all pulses (including various species of cultivated *Vigna*) to varying extents. This generates heterozygosity and brings about substantial heterogeneity in the population, resulting in the loss of newly developed cultivars if they go unnoticed. However, on the other hand, it makes them “resilient” to changing climate conditions, as heterozygosity in the population appears to confer resistance to environmental change [5]. Heterogeneity in plant populations accelerates opportunities for the selection of more stress-tolerant genotypes and thereby provides resilience to the crop as well as the ecosystem [6]. Crop plant resilience, therefore, appears to be brought about in nature by the shuffling and recombination of genes at many loci, leading to the creation of novel adaptive attributes which ultimately result in enhanced “adaptedness” for a few recombinants in the changed environmental condition.

Presently, the impact of global warming can be seen worldwide. For example, India has witnessed highly fluctuating weather conditions in the last decades [7]. It is evident that high temperatures have changed the rainfall pattern as well as distribution and have increased water scarcity. In the future, the shortage of water will increase drought-affected regions. Moreover, it will negatively impact those regions that have higher precipitation rates [8]. The impact of climate change on chemical and physical processes in soils and nutrient uptake from soils has previously been reviewed comprehensively [9]. In Myanmar, erratic rainfall due to climate change had a detrimental impact on pulse production efficiency [10]. Thus, aberrant weather conditions (global warming) are expected to pose serious threats to pulse productivity in the near future as rising temperatures will lead to production of poor biomass; reductions in days to flowering, rate of fertilization, and seed formation [11,12,13,14,15]; and intensifying vulnerability to disease and insect pests [1,16,17]. As per a Food and Agriculture Organization (FAO) report [18], climate change has put global food security more at risk; heightened the dangers of undernutrition in resource-poor regions of the world due to heat, drought, salinity, and waterlogging; and increased the threat of newly emerging diseases and insect pests. While assessing the impact of drought on crop yields, Kuwayama et al. [19] reported 0.1–1.2% yield reduction for corn and soybeans for each additional week of drought. According to Ambachew et al. [20], drought stress can cause 20–90% yield reduction in common bean, which in the worst scenario could go up to 100%. In other pulses, yield losses have been measured to the extent of 6–86% and 15–100% due to different abiotic and biotic stresses, respectively [21]. Although McKersie [8] has discussed a number of options for mitigating the effects of climate change on crop production, breeding for genotypic adaptation is one of the important strategies for dealing with future climate change [22]. It involves incorporating novel traits in crop varieties to enhance food productivity and stability. For breeding climate-resilient cultivars in pulses, it is imperative to bring about genetic improvements for adaptive traits [22,23]. Shunmugam et al. [24] reviewed the physiological traits that may facilitate breeding climate-resilient food legume crops for adaptation under abiotic stresses. The symbiont preference traits related to abiotic stresses have recently been studied in the model legume *Medicago truncatula* [25]. Cullis and Kunert [26] unlocked traits that impart drought tolerance by producing a range of secondary metabolites and proteinaceous inhibitors in response to environmental stresses in orphan legume crops. As climate change is the biggest threat to the production of both warm- and cool-season pulses in the coming years, the mining of adaptive traits in the germplasm to transfer them into newly bred cultivars is highly desirable. Information on this aspect of pulse crops is still scattered in the literature. In this review, we have therefore made an attempt to organize such dispersed information and discuss various strategies to exploit adaptive traits for breeding climate-resilient smart pulses.

## 2. Overview of Adaptive Traits in Pulses

Climate change can result in a wide range of abiotic stresses, such as drought, heat, cold, salinity, flood, and submergence, and biotic stresses, including increased attacks of pathogens and pests [27]. Therefore, breeding of adaptive traits is required for increasing the resilience of crops to current climate change conditions to help sustain productivity. Adaptive traits show their adaptive plasticity in changing environmental conditions and help crop plants survive and/or reproduce under biotic and abiotic stress conditions [26]. These adaptive traits can be agromorphological [28,29], physiological, and biochemical [24,26,30,31]. In pulses, breeders have attempted to improve many traits for the given target environment. Therefore, specific adaptive traits must be incorporated in the improved genotypes for each growing condition (Table 1).

The reproductive stage substantially influences seed yield in crop plants. It has been reported that drought stress during the pod-filling stage leads to pod abortion and thus reduces the number of seeds per plant, whereas terminal drought at the early podding stage resulted in an 85% decline in seed yield of chickpea [32]. Thus, pod-filling ability can be targeted as an agromorphological trait under moisture-deficient conditions for developing drought-resilient cultivars. In Mediterranean environmental conditions, leaf traits such as leaf area, leaf weight, and leaf growth rate have been identified for tolerance to drought stress [33]. Physiological traits, which are relatively stable across environments, provide greater breeding value [34]. A number of physiological traits including leaf parameters, seed set, pod abscisic acid concentrations, and root traits have been shown to impart tolerance to drought in chickpea [32,35]. The role of sucrose infusion has recently been identified in the salt tolerance of chickpea [36]. Prince et al. [37] performed an innovative analysis to decipher the mechanisms that underpin drought tolerance in legumes and established the role of root xylem plasticity in improving water-use efficiency in soybean plants subjected to water stress [37]. An extensive root system is a useful drought-avoiding physiological trait that helps to maintain the seed yield under drought conditions in pulses through enhanced extraction of soil water [38,39]. It is therefore desirable to exploit root and leaf traits while breeding for drought avoidance in pulses.

It has been established that proteins and metabolites are generated in different tissues of crop plants in response to environmental stresses [24]. Identification, quantitation, validation, and characterization for a wide range of proteins/metabolites from specific organ/tissue/cells under stress conditions can be useful biochemical traits for breeding climate-resilient crops. Protein differential expression analysis in response to various stresses at different growth stages has been studied in several pulse crops including chickpea, pea, green gram, and common bean [40]. Various morphophysiological traits imparting tolerance to such stresses have been identified in pulse crops for their wider adaptability considering the global trend of rising temperatures over the years [41,42,43,44,45,46,47]. One of the key physiological traits is photosynthetic activity, as high and low temperatures cause photodamage to photosystem-II (PS-II) [48,49]. In lentil, pollen and leaf traits could be helpful in identification of heat-tolerant genotypes [15]. Breeders have exploited early flowering traits in chickpea breeding programs, leading to the development of new chickpea varieties adapted to warmer, short-season environments that resulted in a chickpea revolution in southern India [50]. For lentil, the development of short-duration cultivars has increased the opportunity for the adaptation of lentil crops in rice-fallow areas owing to reductions in yield losses caused by forced maturity [51]. Adaptation towards freezing temperatures involves a number of structural and functional changes at the cellular level. During acclimation, organic compounds such as sugar, proline, and glycine betaine accumulate in plant cells and confer frost tolerance to surviving plants. One such organic compound, “glycine betaine”, has been shown to mitigate cold stress damage in chickpea [52]. It is therefore obvious that the morphophysiological and biochemical parameters imparting adaptive value vary with the nature and kind of abiotic stress and pulse species, respectively. For breeding smart pulses for a specific situation, special adaptive features need to be exploited.

## 3. Looking Back to Wild Species and Land Races for Adaptive Traits

The growing intensity of abiotic and biotic stresses calls for adoption of mitigation and adaptation strategies by incorporating resistance/tolerance to various stresses to increase resiliency and sustain the productivity of pulse crops in a changing climate scenario. These strategies will pave the way for efficiently meeting humankind’s demand for a more plentiful and nutritious food supply [93,94,95,96,97,98,99]. Cultivated species of pulses have narrow genetic diversity to withstand current global warming challenges [100]. It is therefore necessary to look back to wild species and land races when searching for useful adaptive traits/genes. It is well documented that wild species have a reservoir of many useful genes [47,101] because they have evolved under natural selection to survive climatic extremes and can potentially provide further genetic gains [93,94,96]. Therefore, wild species need to be exploited in genetic improvement programs to alleviate the challenges of global warming and its related effects on pulses.

Wild relatives of crops have been used sparingly and typically in an ad hoc manner in many crop breeding programs [96,102,103]. In pulse crops, Sharma et al. [101] reported a number of wild accessions having high levels of resistance/tolerance to various stresses [101]. As wild relatives of chickpea and lentil are native to drought-prone areas, they possess useful traits for drought tolerance. According to Gorim et al. [104], an evaluation of wild relatives of lentil for root and shoot traits under water-deficit and fully watered conditions resulted in different patterns of root distribution into different soil horizons. The study revealed that wild lentil genotypes employed diverse strategies such as delayed flowering, reduced transpiration rates, reduced plant height, and deep root systems to either escape, evade, or tolerate drought conditions.

The use of wild species for targeted introgression of useful genes dates back to the work of Vavilov [105]. Thereafter, crop wild relatives have been used continually to transfer adaptive traits in a variety of crops including pulses. According to Maxted and Kell [96], more than 291 articles have come out on pulses regarding the identification and introgression of useful traits from 185 wild relative taxa to 29 crop species. Most of these studies have focused on disease and pest resistance (>50%), abiotic stress tolerance (10–15%), and yield traits (20%). Also, 74% of 104 molecular-assisted breeding studies (1995–2012) have dealt with introgression of traits from wild species that confer disease resistance, while the remaining studies covered abiotic stress tolerance, improved yield, and growth habit [103]. Thus far, resistance to many diseases and insect pests from wild relatives and unadapted germplasm has been successfully transferred into suitable genetic backgrounds [102,106,107,108,109]. Kumar et al. [109] recorded useful genetic variability for days to 50% flowering, secondary branches, number of pods/plant, biological yield/plant, grain yield/plant, and 100 seed weight in the indigenous gene pool of lentil. *Lens ervoides* (a wild species of cultivated lentil) has been exploited in Canada for transferring anthracnose resistance genes into cultivated backgrounds through embryo rescue techniques [110,111]. More recently, the crossing of cultivated species with *Lens tomentosus* accession “ILWL120” followed by ovule culture has resulted in the development of a number of prebreeding lines carrying diversity for flower color, seed coat, and cotyledon color [112]. In India, under an ICAR-ICARDA network project, many prebreeding lines (>500) developed by using crossable wild species of lentil have shown variability for yield-contributing traits. In chickpea and pigeonpea, wild relatives have been exploited for enhancing the adaptability of cultivated species against climate extremes under changing climate conditions [47,101]. Brumlop et al. [113] developed from a *C. arietinum* × *C. judaicum* cross the prebreeding line “IPC 71”, which has an increased number of primary branches, pods per plant, and green seeds for further use in chickpea improvement programs. These reports and achievements substantiate the fact that wild species of pulses do carry potentially useful genes. Considering current climate variability and its manifold effects [101,114], such wild species need to be exploited for developing prebreeding lines of pulses for the new environment. Their utilization in pulse breeding programs may result in climate-resilient smart cultivars with a broad genetic base and the ability to sustain environmental extremes [101].

## 4. Conventional Breeding Approaches

Conventional breeding approaches have been used to tailor suitable plant types with the ability to adapt to different environmental niches/cropping systems. To this end, breeders focused mainly on highly heritable visually adaptive traits (agromorphological traits). The use of dwarfness and leaflessness (i.e., modification of leaflets into tendrils) traits in field pea resulted in the development of new plant types that allow penetration of sunlight to lower portions of the plant, provide natural mechanical support to preclude lodging, and prevent bird damage owing to a network of interlocked tendrils above the crop canopy [115]. Recently, Saxena et al. [116] recommended an ideal plant type of pigeonpea comprising rapid seedling growth; nondeterminate (NDT) growth habit; spreading or semispreading branches; a greater number of secondary and tertiary branches; long fruiting branches; more flower bunches; 5–6 pods/bunch; 4–6 seeds/pod; 12–14 g/100 seed weight; resistance to *Fusarium* wilt (FW), sterility mosaic (SM), and *Phytophthora* stem blight (PSB); deep root system/drought tolerance; and the ability to mitigate other abiotic stresses including waterlogging for pigeonpea–cereal intercropping systems. An early flowering exotic line “Precoz” (ILL 4605) of lentil has been utilized extensively to tailor plant architecture having vigorous growth, medium maturity, large seeds, and cold tolerance, particularly for Indo-Gangetic plains [109]. Earliness, which provides an escape mechanism from drought and terminal heat stresses, has been invariably used in almost all breeding programs to mitigate such stresses [1,117]. In chickpea, significant progress has been made in developing early maturing varieties that mature in 85–90 days in peninsular India [118]. Even extra short duration chickpea varieties, termed super-early types, have been reported in chickpea [119] and pigeonpea [120], and efforts in this direction are also underway in other pulse crops. Some super-early lines maturing within 100 days with a yield potential up to 1.5 t/ha have been reported by International Crops Research Institute for the Semi Arid Tropics (ICRISAT) in both determinate (DT) and NDT groups of pigeonpea [120]. For sustaining crop intensification under the rice-fallow system of eastern India, development of an early maturing variety (90–100 days) has been suggested in lentil, and efforts are being made to tailor genotypes having earliness and high biomass and harvest index [12]. Kashiwagi et al. [117] identified root traits to improve water uptake in chickpea under limited-moisture conditions. They used contrasting chickpea accessions vis-à-vis root biomass and rooting depth in a drought-avoidance breeding program to improve the root system of ensuing genotypes for cultivation in central and southern India.

In the coming years, the disease and pest scenario may be a serious problem due to climate change. Therefore, climate-smart pulses must carry resistance to diseases and insect pests. Germplasm screening under natural and artificial conditions to identify resistant sources for various diseases and insect pests has been a regular feature of resistance breeding programs for pulses [2,121]. Knowledge of the genetics of resistance traits and racial composition of pathogens has accelerated the development of cultivars having adaptability under epidemic conditions [2,122,123]. Recently, root rots (dry and black root rots) and collar rots have emerged as potentially damaging diseases in both chickpea and lentil. However, the literature pertaining to the racial description of the causal organisms (species of *Rhizoctonia*, *Fusarium*, and *Sclerotinia*) and resistant donors in both these crops is still scanty. These diseases need to be tackled through breeding in the days ahead. However, the complex nature of these diseases, the resistance mechanisms and traits (especially physiobiochemical traits), and the limited screening facilities are the major limitations to progress for pulses through conventional plant breeding. These limitations need to be overcome through phenomics-based breeding, which is currently employed for improving soybean [124].

## 5. Omics-Based Breeding Approaches for Adaptive Traits in Pulses

Different “omics” fields, namely, genomics, transcriptomics, epigenomics, proteomics, metabolomics, and phenomics, have emerged during the past years. These approaches have enhanced the precision and sped up the ongoing breeding programs of the major food crops such as wheat and rice [125]. Omics-based strategies can also be used to develop climate-smart pulses (Figure 1). These strategies have been categorized as current and emerging omics-based approaches and are discussed below.

### 5.1. Current Genomics Approaches

During the last 25 years, substantial advances have been made in the genomic resources of pulse crops, leading to the development of various molecular markers and the availability of QTLs/genes that impart tolerance to various biotic and abiotic stresses. These genomic resources have been discussed earlier in detail [126,127]. Next-generation sequencing (NGS)-based genomics tools have enabled rapid and cost-effective identification of the functional and regulatory genes controlling abiotic stress resistance in many pulses [128]. These NGS tools have helped to develop SNP and INDEL markers [129] and expression atlases [130,131,132] and to understand the signaling pathways for tolerance to various environmental stresses in various legumes including pulses [128]. Genome sequences of major pulses, which are important genomic resources for translating the genomics into the field are now available in the public domain [133,134,135,136]. Genomics approaches have now become an integral part of the current conventional breeding program and can be used in different ways for shortening the period of genetic improvement and targeted manipulation of genomes for climate-smart pulses.

#### 5.1.1. Molecular Markers Associated with Adaptive Traits in Pulses

Knowledge of genes controlling traits for wider adaptability is the prerequisite to develop climate-smart pulses. In the last three decades, efforts have been made to identify such genes/QTLs (Table 2) in chickpea, pigeonpea, and other pulse crops. In these crops, QTLs/genes for pods per plant (qPD4.1) and flowering (qFL4.1 and qFL5.1) in pigeonpea [137] and thermotolerance [138] in chickpea can help to construct new plant types suitable to changing environments [13]. Moreover, QTLs (*HQTL-1* and *HQTL-2*) have been identified for pollen viability in azuki bean [139]. In field pea, Javid et al. [140] validated markers associated with abiotic and biotic stresses for breeding programs. In this study, a molecular marker “PsMlo” showed an association with powdery mildew (PM) resistance and boron (B) tolerance, while several other markers were found associated with salinity tolerance across a diverse set of pea germplasm. The PsMlo1 marker predicted the PM and B phenotypic responses with high levels of accuracy (>80%) and thus showed its potential to facilitate improvement for PM resistance and B tolerance. More recently, Paul et al. [141] identified QTLs associated with heat tolerance in chickpea in a mapping population comprising recombinant inbred lines (RILs) evaluated under two heat-stress (late sown) and one nonstress (normal sown) environments. This resulted in the identification of 25 putative candidate genes responsible for heat stress in the two major genomic regions. The identified markers, which were linked to four major QTLs, can be utilized in breeding programs. For improving the adaptation of common bean to adverse environments, Diaz et al. [142] evaluated RILs under different abiotic stress conditions for a number of agrophysiological traits and identified molecular markers linked with QTLs for abiotic stress tolerance. In accessions of common bean from the northwestern Himalayas, Choudhary et al. [143] discovered the gene/QTLs for *Anthracnose*. These genes/QTLs need to be utilized in breeding programs for developing stress-tolerant cultivars of common bean.

Molecular marker technology helps plant breeders and gene bank curators to identify markers linked with morphological and physiological traits in the available germplasm that enhance crop adaptation under climate variability [170]. According to Kumar et al. [170], these linked markers can be used to develop “climate change ready” cultivars for cultivation. The application of molecular marker technology has resulted in the identification of markers linked to genes controlling several abiotic and biotic stresses [171,172,173,174,175] and other agronomic traits [53,174,176,177,178,179,180] in chickpea and pigeonpea. Also, the draft genome sequence of both Kabuli and Desi chickpeas and pigeonpea are available in the public domain [134,181,182]. These sequence data of chickpea and pigeonpea will assist in enhancing their productivity and lead to conserving food security in arid and semiarid environments. Many QTLs have been identified on several linkage groups (2, 3, 4, 6, and 8) for *Aschochyta* blight (AB) resistance [183], and marker-assisted backcrossing (MABC) has been used for conversion of targeted lines with respect to one or two traits without disturbing other native traits of the target variety in chickpea [184]. According to Varshney et al. [183], simultaneous genetic improvement for FW and AB resistance is possible through marker-assisted selection (MAS) in chickpea. To this end, they undertook two parallel MABC programs by targeting the *foc 1* locus and two QTL regions, namely, ABQTL-I and ABQTL-II to introgress resistance to FW and AB, respectively, in “C 214”, an elite cultivar of chickpea. Phenotyping of lines developed through MAS led to the identification of some lines carrying both FW (race 1) and AB, resistance which would be tested further for yield and other agronomic traits under multilocation trials for possible release and cultivation. In an attempt to identify QTLs for root traits in chickpea, Serraj et al. [185] developed a RIL population from a cross between a long root genotype “ICC 4958” and a well-adapted, high yielding variety “Annigeri”. This RIL population was used to map the genes/QTLs for root traits, leading to the identification of a “QTL hotspot” that explained a large part of the phenotypic variation for major drought tolerance traits, including the root traits. Kashiwagi et al. [117] used marker-assisted breeding to introgress this QTL hotspot into a leading Indian chickpea cultivar “JG 11”. They demonstrated that introgression lines had shown a distinct yield advantage (>10%) over JG 11 in multilocation evaluations under terminal drought. These marker-based success stories of chickpea can also be replicated for improving stress tolerance in other pulse crops. Choudhary and co-workers [186,187] used root traits to screen pigeonpea genotypes against Al toxicity and established root exclusion as the possible mechanism for Al tolerance, whereas Daspute et al. [188] discovered Al-responsive citrate excretion as the biochemical basis of Al tolerance in pigeonpea. The information generated in pigeonpea may be utilized in other pulses for improvement of Al tolerance.

#### 5.1.2. Gene(s) Related to Adaptive Traits

Since climate changes have a large influence on the creation of a number of biotic stresses, knowledge of genes that express themselves in different environmental conditions can help in breeding climate-resilient crops. Transcriptome analysis has helped to deliver functionally associated gene-based markers for breeding activities in lentil, field pea, and faba bean [189,190]. In several pulses (pea, lentil, chickpea, common bean, pigeonpea, and broad bean), transcriptomic data have been generated that can be used in gene-based marker discovery to assess genetic diversity, linkage mapping, and trait dissection [191]. In other crops, it has been widely used to identify the candidate genes that express themselves in specific environmental conditions (heat stress) or at particular plant growth stages [192]. Therefore, similar strategies can be employed for identification of such genes/traits imparting wider adaptation to pulse crops under global warming conditions. In chickpea, this approach has been used to identify candidate genes governing plant height and agromorphological traits [193,194], and in lentil, QTLs for B toxicity tolerance, flowering time, and seed characteristics [195,196]. Similar efforts have led to the identification of heat-responsive genes that are expressed in heat-sensitive and heat-tolerant genotypes during heat stress in chickpea [132,197]. In other legume crops, genes responsible for thermotolerance in soybean (*GmHsfA1*) and broad bean (*VfHsp17.9-CII*) have been cloned [198,199]. Naser and Shani [200] mentioned the importance of auxin-related genes that play an important role in plant growth, seed development, and abiotic stress response (drought and salinity tolerance). In pigeonpea, Pazhamala et al. [131] developed a compendium of 28,793 genes that express themselves during the reproductive stage in seed-forming tissues and identified a network of 28 flower-related genes. Similarly, Singh et al. [201] established a transcription factor database (i.e., PpTFDB) in pigeonpea that can be useful for functional genomic analysis in other legume crops [201]. Kudapa et al. [202] developed a comprehensive gene expression atlas by associating genome sequence with genes expressed across different plant developmental stages and organs covering the entire lifecycle of chickpea. They identified 15,947 unique numbers of differentially expressed genes and observed significant differences in gene expression patterns in the process of flowering, nodulation, and seed and root development. They could also identify candidate genes responsible for drought stress from the QTL hotspot region. These recent advances, including the development of gene expression atlases and signaling pathways involved in plants’ responses to environmental stresses, will certainly facilitate the development of climate-smart pulses.

RNA-sequencing-based (NGS-based) transcriptome analysis is considered to be a superior approach to understand the gene function and molecular basis of many cellular responses in plants exposed to abiotic stresses. The gene expression analysis performed by Abdelrahman et al. [128] could identify a number of candidate genes for drought, salinity, cold, and heavy metal stress resistance in chickpea and other pulses. According to Singh et al. [203], transcriptome changes occur in response to seedling drought stress in lentil. They recognized the upregulation of genes involved in electron transport chains, oxidation-reduction processes, the TCA cycle, senescence and reduction of stomatal conductance, the downregulation of genes associated with gamma-aminobutyric acid synthesis, transcription binding and synthesis of cell wall proteins, and the negative regulation of abscisic acid responses in the drought-tolerant lentil genotype “PDL 2”. Studies on the MLO gene family [145,204] have revealed that the *LcMLO1* and *LcMLO3* genes in lentil and the *PsMLO1* gene in pea are associated with PM resistance. In chickpea, Garg et al. [205] carried out a comparative transcriptome analysis of drought- and salinity-tolerant/sensitive genotypes at different developmental stages. They could identify genes encoding enzymes involved in the biosynthesis of sugar alcohols (inositol and trehalose), xyloglucan, and amino acids (proline and citrulline). The results of the transcriptome study represented a starting point to dissect the gene regulatory networks involved in drought and/or salinity stress in chickpea [205]. Transcriptome analysis performed in the nodules involving *Mesorhizobium ciceri* CP-31-(McCP-31)-chickpea and *M. mediterraneum* SWRI9-(MmSWRI9)-chickpea associations under P_i_-deficient and -sufficient conditions could identify changes in the expression of genes in more-P_i_-deficiency-sensitive MmSWRI9-induced nodules than in less-P_i_-deficiency-sensitive McCP-31-induced nodules [206]. Recently, Mashaki et al. [169] studied transcriptome profiles in roots and shoots of two contrasting Iranian kabuli chickpea genotypes under water-limited conditions at the early flowering stage using an RNA-sequencing approach. They identified 4572 differentially expressed genes (DEGs) and grouped these DEGs into several subcategories depending upon the intensity of drought stress. Also, several transcription factors (TFs) controlling major metabolic pathways such as ABA, proline, and flavonoid biosynthesis have been identified, spotting DEGs in QTL hotspot regions (reported earlier) in chickpea. Thus, genes/TFs upregulated in the drought-tolerant genotype during drought stress in this study are potential candidates for enhancing tolerance to drought [169]. Moreover, an early flowering1 (Efl1) gene, which is an ortholog of the early flowering3 (ELF3) gene of *Arabidopsis* (*Arabidopsis thaliana*), has also been mapped and sequenced in chickpea [207]. It is therefore expected that integration of phenomics with transcriptomics, proteomics, and metabolomics will provide greater insight into the molecular changes occurring during the growth and development of various species of pulses under environmental stresses.

#### 5.1.3. Transgenics for Increasing Adaptability of Pulses

Gram pod borer (*Helicoverpa armigera* Hubner) is the key insect pest of pigeonpea and chickpea, causing 17–35% yield losses [208]. No resistance sources to this insect pest are available in the cultivated germplasm and immediate wild progenitors of these two pulse crops. According to Choudhary et al. [121], wild relatives of pigeonpea, notably *Cajanus scaraeboides* and *C. platycarpus*, have morphologically adaptive features that impart resistance to pod borer. The resistance-imparting morphological traits in such wild species include density of nonglandular trichome *C* on pods (>5 times greater than that present on pods of cultivated accessions), width and waxiness of pod wall, and prominent pod constrictions. Attempts to develop pod-borer-resistant genotypes of pigeonpea and chickpea by conventional breeding methods have not been very successful due to crossable barriers and incompatibility with wild species. Moreover, the incomplete penetrance and variable expressivity of such wild genes in the cultivated background further complicates the outcome [121]. Therefore, a transgenic approach has been adopted to improve resistance to pod borer in both pigeonpea and chickpea. This has resulted in the development of transgenic lines of pigeonpea and chickpea carrying Bt genes, namely, *cry*1Ac, *cry*1Ab, and *cry*2Aa. The transgenic lines of chickpea with synthetic Bt genes either singly or in combination have exhibited a high level (98–100%) of mortality of *Helicoverpa* larvae [209,210]. Das et al. [211] reported that field trials of several such transgenic lines are underway at the Indian Institute of Pulses Research (IIPR), Kanpur. Efforts have also been made to utilize some of these effective lines in backcross breeding program for further improvements [211]. According to Singh et al. [212], transgenic plants expressing the Cry2Aa gene have been developed employing *Agrobacterium*-mediated in planta transformation approach in pigeonpea. Developed transgenic plants (T_3_ lines) have demonstrated 80–100% mortality of the challenged larvae and improved the ability to prevent damage caused by the larvae. The selected transgenic plants accumulated Cry2Aa in the range of 25–80 µg/g [212]. Transgenic approach has also been used to tackle the problem of salt tolerance in chickpea and pigeonpea [213,214]. According to Bhatnagar-Mathur et al. [214], the osmoregulatory gene “*P5CSF129A*”, encoding overproduction of proline transferred through genetic transformation, confers drought tolerance in chickpea. It is thus obvious that the ongoing efforts to develop effective transgenic lines for biotic and abiotic stresses will yield desired results very soon in both chickpea and pigeonpea.

### 5.2. Emerging Omics Approaches for Breeding of Adaptive Traits

Though knowledge of epigenomics, proteomics, metabolomics, and genome editing is still limited in pulses, these approaches have opened up new avenues for resolving the complexity of adaptive traits imparting tolerance to biotic and abiotic stresses. These diverse omics platforms have great potential for improving the current understanding of important traits, enabling us to develop new strategies for developing climate-smart pulses.

#### 5.2.1. Integration of Proteomics and Metabolomics with Genomics for Enhancing Climate Resilience

Proteomics and metabolomics have emerged as cutting-edge areas of functional biology [40]. Integration of information obtained from proteomics and metabolomics with genomics data can enhance our understanding about plants’ response to abiotic stresses [215]. Several studies have revealed a network of stress responsive genes/proteins/metabolites/transcription factors in various legumes, including soybean [215,216,217]. This can help to catalogue and prioritize the genes to exercise selection of superior traits for realizing genetic gains in crop breeding programs [218]. Recent advances in proteomics have included classification of proteins, comparison of protein profiles, post-translational modifications of proteins, identification of protein complexes and interacting networks, study of protein structure and functional groups, and their use in crop improvement [219].

In legumes, proteomic studies have unraveled the molecular mechanisms underlying tolerance to different biotic (AB, PM, FW, rust, mungbean yellow mosaic India virus, aphids, etc.) and abiotic (drought, waterlogging, salinity, cold, heat, mineral deficiency, heavy metal toxicity, and dark and UV–B irradiation) stresses [220,221]. After performing a proteomic analysis, Krishnan et al. [222] identified 373 proteins in pigeonpea seeds. They observed a large number of seed proteins showing significant homology for amino acid sequences with that of soybean seed proteins. They could recognize a large number of stress-related proteins which probably confer adaption to pigeonpea in drought-prone environments. More recently, Rathi et al. [223] identified and characterized proteins that enhance adaptation of grass pea under dehydration conditions. Based on their putative functions, they grouped these proteins into 22 functional categories, and 9.17% of these proteins showed their relation with dehydration-induced stress [223]. In faba bean, Li et al. [224] carried out leaf proteomic analysis under drought stress. They could classify quantified proteins mainly into five functional groups (regulatory proteins: 46.7%; energy metabolism: 23.3%; cell cytoskeleton: 6.7%; other functions: 20%; and unknown function: 3.3%). This study showed upregulation of chitinase, 50S ribosomal protein, Bet protein, and glutamate–glyoxylate amino-transferase under drought conditions, suggesting their important roles in drought tolerance [224]. According to Lin et al. [225], integration of transcriptomic and proteomic research has been fruitful for exploring bruchid-resistant genes in mungbean. This study will have a far-reaching impact on the control of bruchid (Callosobruchus spp.), which infests grains of almost all pulses in storage.

Various metabolic changes occur in plants when they are exposed to abiotic stresses [226]. Knowledge of metabolite profiles provides insight into the functional role of metabolites for traits imparting tolerance/resistance to abiotic and biotic stresses. In addition, integration of gene expression profiles with metabolite profiles helps to identify gene-to-metabolite associations/networks [40,227]. A few studies have been conducted to determine the metabolic profiles of legume crops, including pulses. Metabolic changes that take place during legume–rhizobial symbiosis have been studied under different conditions, including drought stress [228,229,230,231,232,233]. In common bean, Hernández et al. [229], after analyzing the nontargeted metabolite profile, identified changes in the roots and nodules of plants inoculated with *Rhizobium tropici* grown under P_i_-deficient and -sufficient conditions. In this study, metabolic differences were observed between plants grown under these two contrasting conditions. Thirteen metabolites showed their role in those pathways that repressed or induced pathways in response to P_i_ deficiency. Nodules of P_i_-deficient common bean plants showed a reduction in N-metabolism-related metabolites, which might contribute to a decrease in symbiotic nitrogen fixation (SNF) efficiency. In lentil, metabolite profiling of four different Mediterranean accessions performed by Muscolo et al. [234] showed that intermediates of the TCA cycle and glycolytic pathways decreased under drought and salinity conditions. Moreover, they recognized stress-specific metabolites such as threonate for NaCl and asparagine/ornithine and alanine/homoserine specifically to drought and salinity, respectively. In chickpea, Nasr Esfahani et al. [233] observed significant differences in C- and N-metabolism-related metabolites in the more-P_i_-deficiency-susceptible *Mm*SWRI9-chickpea nodules and the less-P_i_-deficiency-susceptible *Mc*CP-31-chickpea nodules under P_i_ deficiency. Moreover, they noted a remarkable increase in the level of organic acids in *Mc*CP-31-nodulated roots as compared with *Mm*SWRI9-nodulated roots under P_i_ deficiency. This study showed that a crosstalk among various signaling pathways involved in the regulation of *Mesorhizobium* chickpea exists for adaptation to P_i_ deficiency. Further in-depth knowledge at the genetic level can be useful for developing transgenic cultivars in leguminous crops to have adaptability under P_i_ deficiency by sustaining efficient SNF. In recent years, whole-genome sequences, genome-wide genetic variants, and cost-effective genotyping assays have emerged and provided an opportunity to utilize metabolomics information for the genetic enhancement of adaptive traits towards the ultimate aim of developing climate-resilient pulses [235].

#### 5.2.2. Epigenomics for Improving Phenotypic Plasticity to Climate Change

Epigenetics denotes heritable changes in gene expression that can occur due to methylation of DNA or post-translational modification of histones involved in chromatin formation rather than changes in gene sequences [236]. Epigenetic variations can be reversible or transgenerational [237]. In reversible epigenetic variation, transcriptional memory may be responsive to cell fate decisions, developmental switches, or stress responses; otherwise, the gene expresses itself normally. Such epigenetic variations are not inherited by the next generation and, hence, are not useful for epigenetic breeding. Iwasaki [238] discussed the chromatin resetting mechanism related to the stability of epigenetic states under various stress conditions and revealed that silencing of reporter transgenes as well as endogenous loci occurs in response to various abiotic stresses such as high salinity, drought, heat, or UV radiation. However, such stress-induced transcriptional activation is mostly transient, and silencing is rapidly restored after resumption of optimal growth conditions. On the other hand, transgenerational epigenetic variation causes changes in gene expression that are stably transmitted to subsequent generations through mitosis or meiosis. Such epigenetic marks that create natural phenotypic variation play an important role in adaptations of plants in different environmental conditions [239,240,241,242]. Epialleles or epimutations, creating a mutant or an alternative phenotype, are generated through epigenetic changes [237,243]. According to Slotkin and Martienssen [244], these epialleles may result from changes in either genome or environmental conditions. Zhang and Hsieh [245] identified pure epialleles originating independently of any genetic variation in their model and other crop plant species. However, epialleles which are caused by genetic variations are difficult to detect without comprehensive genome structural analysis. Therefore, it is challenging to identify genomic loci that undergo epigenetic changes in response to environmental conditions [246]. According to Meyer [246], the use of such genomic loci in epigenetic breeding is a powerful strategy for developing climate-smart pulses under global warming conditions because such epialleles are able to improve the plant’s ability to adapt to the inducing conditions in a heritable manner. Lele et al. [242] used amplified fragment length polymorphism (AFLP) and methylation-sensitive AFLP (MSAFLP) to identify the differences in genetic diversity caused by epigenetic or genetic variations and to study the role of epigenetic variation in the adaptation of *Vitex negundo* var. *heterophylla* (Chinese chaste tree) in different habitats. This study showed a relatively high level of genetic and epigenetic diversity but very low genetic and epigenetic differences between habitats within sites.

DNA methylation, a well-known form of epigenetic modification in plants, regulates genomic imprinting, expression of genes, and the process of disease development in plant species [247]. It influences transcription activity, morphological development, agronomic trait formation, the process of disease development, and environmental adaptation [247,248]. However, technological advancements have made it feasible to identify methylomes at a single-base resolution using BS-seq in soybean [248]. Methylome profiles studied among diverse accessions in key crop species including chickpea and soybean showed thousands of differentially methylated regions (DMRs) [248,249,250]. Genome-wide cytosine methylation analysis has been done on soybean for roots, stems, leaves, and cotyledons of developing seeds at single-base resolution. This study identified 2162 differentially methylated and hypomethylated regions, which provided significant insight into soybean gene expression [251]. Other studies have identified the role of DNA methylation in controlling cytoplasmic male sterility [252] and seed development in soybean [253]. Moreover, it has also played an important role in the polyploidy of soybean and common bean [254]. A whole-genome DNA methylation investigation performed by Shen et al. [248] identified 5412 DMRs which are useful in the domestication and improvement of soybean. The study also identified DMR-enriched genes belonging to carbohydrate metabolism [248]. In tetraploid cotton (a nonlegume crop), Song et al. [255] recognized 519 genes differing epigenetically between wild and cultivated species. Among these genes, a few methylated genes were responsible for traits such as flowering time and seed dormancy which helped in the domestication of cotton. This study also showed that DNA methylation changes the expression of the genes of wild species in response to environmental conditions or during the human selection. This study further revealed that the methylated gene helped cotton to adapt in natural tropical environments because DNA methylation of this gene does not encourage flowering under long-day conditions. Bhatia et al. [250] also identified DMR-associated genes involved in the development of the flower of chickpea. In addition to this, natural variation of epialleles has provided an opportunity for plant breeders to select and breed agronomically important traits [256,257]. As pulses have undergone domestication under varied agroclimatic conditions, a comprehensive investigation of methylated genes among accessions belonging to wild and cultivated species will help identify DMR-enriched genes that might affect their adaptedness to local climatic condition.

A breeding strategy was suggested by Raju et al. [258] to exploit epigenetic variations for increasing yield and stability in soybean. This strategy employed the MutS HOMOLOG1 (MSH1) system to induce epigenetic variation for agronomic traits. For epigenetic breeding, epi-lines were developed by crossing between wild type and *msh1-*acquired soybean memory lines, which showed a wide variation for multiple yield-related traits including pods per plant, seed weight, and maturity time in both greenhouse and field trials. Low extent of epitype-by-environment (e × E) interaction indicated higher yield stability. Furthermore, transcript profiling of the soybean epi-lines helped to identify genes involved in various metabolic pathways responsible for enhanced growth behavior across generations. This indicated the potentiality of MSH1-based epigenetic variation in plant breeding for enhanced yield and yield stability [258]. Thus, environmentally induced epigenetic variation can result in heritable phenotypic plasticity, which may play a major role in adaptation to environmental change [239,258,259]. Since pulses are grown in a wide range of environmental conditions and face many stresses throughout their lifecycle, breeding for epigenetic variations can be more useful for the ultimate aim of developing climate-smart pulse crops (Figure 2).

#### 5.2.3. Genome Editing Approaches for Adaptation

Genome editing has emerged as a new approach to bring about genetic changes at targeted regions of the genome and is being utilized as an alternative to classical plant breeding and the transgenic approach [260,261]. Genome editing includes insertion, removal, or replacement of a targeted gene. CRISPR/Cas9-based genome editing, used first in 2013 with *Arabidopsis* protoplasts and tobacco cells [262], is a highly advanced system and user-friendly tool for targeted gene manipulation in many plant species, including crop plants [263,264]. Genome editing could induce mutations in targeted genes with a frequency of 1.8–2.7% [265,266]. Among food crops, this approach was used for the first time in rice and wheat [267]. Initially, the rice *PDS* gene (*OsPDS*) was targeted with two sgRNAs (SP1 and SP2), which resulted in a 5% mutagenesis rate in protoplasts. Subsequently, three more rice genes (*OsBADH2*, *Os02g23823*, and *OsMPK2*) and one wheat gene (*TaMLO*) were targeted and mutated in protoplasts [267]. Recently, the CRISPR/Cas9 technology has been used successfully to edit five pyrabactin resistance 1-like (PYL) genes in rice. The mutants generated from the editing of *pyl1/4/6* exhibited the best growth and improved grain productivity up to 30% in natural paddy field conditions [266]. Such mutations (populations) can have better adaptive value in changing environmental conditions. The CRISPR/Cas9 system will likely be a promising alternative to conventional transgenic and breeding approaches that can deliver good results in this field. This system (CRISPR/Cas9) can also be useful for precise manipulation of genes governing adaptation of pulse crops to adverse environmental conditions.

Genetic transformation is now a routine activity in major pulses such as chickpea and pigeonpea, where transgenic plants have already been developed for insect resistance [209,210,212]. Therefore, gene families regulating the ABA pathway that play an important role under abiotic stress conditions identified earlier in legume crops may be targeted for gene editing in pulse crops to tackle the ill effects of the changing climate scenario [267,268,269,270]. The role of the MLO gene family has been identified in controlling powdery mildew resistance, and hence, this gene family may be used in pea and other pulses where powdery mildew is a serious problem [145]. Because the development of cultivars having resistance to pod borer in chickpea and pigeonpea is a major challenge due to the unavailability of resistance genetic resources in the gene pool, gene editing can target genes controlling susceptibility in the host plant of pulse crops as it is used to enhance resistance to viral diseases in plants [271]. Moreover, Wang et al. [272] applied genome editing to understand the basic mechanisms underpinning legume–rhizobia interactions. In several pulses, candidate genes imparting tolerance to abiotic and biotic stress as well as other agronomic traits have been identified [51,138,193,273,274,275]. The gene editing approach can be used to validate the function of these genes, as candidate genes controlling quantitative variations in nodulation have been validated using genome editing [276]. Identified mutant populations can also be useful as genetic resources for breeding improved cultivars and will help strengthen food security in the future.

## 6. Concluding Remarks

For food and nutritional security, it is essential to adopt mitigation and adaptation strategies for sustaining the production and productivity of pulses under changing climate conditions. However, pulse farmers, especially in South Asia and Africa, are poor in resources; hence, they have a limited capacity to adopt mitigation strategies. Consequently, we shall have to resolve the issues of climate change primarily through adaptation strategies. This calls for developing cultivars that can sustain food production in the future. During the past years, many adaptive traits have been targeted knowingly or unknowingly in plant breeding programs. However, breeding climate-resilient cultivars must address moving targets that differ across geographical locations [277,278]. This will help minimize the adverse impact of climate change on agriculture. In addition, we should lay more focus on the use of wild species and land races to enhance crop resilience through evolutionary breeding [279,280]. We should use modern science to bring back diversity in farmers’ fields by developing an evolutionary population (EP) using a mixture of different genotypes of the same crop. As the genetic composition of an EP fluctuates year after year, genotypes having high adaptive value subsequently become predominant in stressful environments [280]. In common bean, such populations are currently grown [281], and farmers claim high yields under stressful conditions [282].

During the last three decades, considerable advances have been made in the genomics of pulse crops, and genome sequences of many pulse crops are now available in the public domain. This has resulted in the identification of genes/QTLs controlling various agromorphological traits. These advances have allowed breeders to incorporate multiple traits into an improved genetic background through genomics-assisted selection, thereby resulting in the development of stress-resilient pulse crops. For example, introgressed lines of soybean carrying the *Ncl* gene have the potential to regulate transport and accumulation of Na^+^ and Cl^−^. This has resulted in 3.6–5.5-fold greater yield advantages over conventional cultivars under salinity conditions. Such advances have made it possible to grow soybean in saline-affected areas [283]. Moreover, introgression of genes from tepary bean (*Phaseolus acutifolius* A. Gray) resulted in the development of elite common bean lines that are able to grow at 4 °C above the limit (18–19 °C) normally tolerated by this crop [284]. In chickpea, efforts to introgress the drought-tolerant QTL into the background of popular cultivars of Africa and Asia through marker-assisted selection have resulted in several chickpea introgression lines. In rainfed yield trials, these lines have shown at least a 10% yield advantage over the recurrent parent [285]. Genome editing technology based on CRISPR/Cas9 can be used to manipulate genes responsible for adaptation in adverse environmental conditions, and the resulting mutant populations can be screened under stressful conditions. Therefore, initiatives for developing climate-resilient varieties using genomics-based approaches merit special attention.

Environmentally induced epigenetic variation has been reported to play an important role in enhancing phenotypic plasticity to changing environments [286,287,288,289,290,291]. Though epigenetic variation has been studied and exploited in other crops, perhaps no reports are available on its use for the improvement of pulse crops. As pulses are grown across a wide range of environmental conditions, concerted efforts are required to study epigenetic variations in these crops. Such efforts may pave the way for climate-resilient smart pulses in the days ahead.

## Figures and Tables

**Figure 1 ijms-20-02971-f001:**
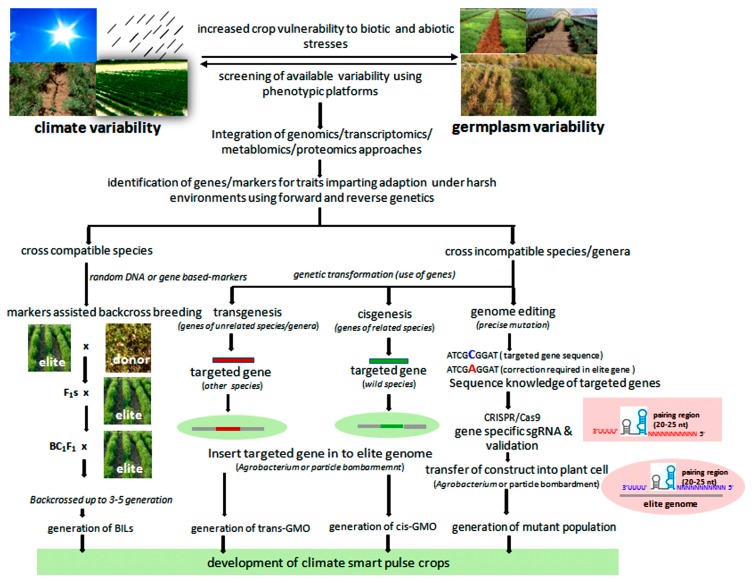
Omics-based approaches for the development of climate-smart pulse crops.

**Figure 2 ijms-20-02971-f002:**
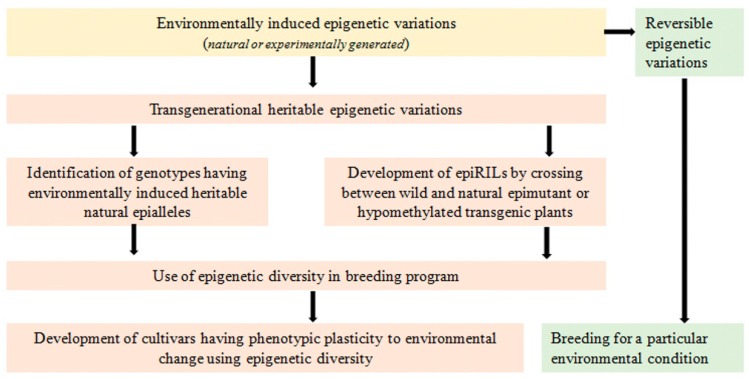
Epigenetic breeding for improving phenotypic plasticity to climate change in pulses.

**Table 1 ijms-20-02971-t001:** Adaptive traits for different growing regions of important pulse crops.

Common/Scientific Name	Region	Adaptive Traits	Reference
Chickpea (*Cicer arietinum* L.)	Nontropical dry areas and semiarid tropics	Earliness; early vigor; spreading to erect growth habit; resistance to pod borer, AB, BGM, wilt, and root rot; tolerance to drought and heat; suitability for mechanical harvesting; herbicide tolerance	[53,54,55,56,57,58,59,60]
Lentil (*Lens culinaris* Medik.)	Nontropical dry areas and semiarid tropics	Earliness; early vigor; spreading to erect growth habit; resistance to wilt, root rot, *Stemphylium* blight, AB, rust, and black aphid; tolerance to drought and heat	[12,13,61,62]
Pea (*Pisum arvense* L.)	Cool, semiarid climates	Dwarfness, leaflessness, tendril, resistance to rust and powdery mildew, tolerance to terminal heat and drought, earliness	[63]
Mungbean (*Vigna radiata* Wilczek)	Arid and semiarid regions, wide adaptation, warm season	Short duration, MYMV and powdery mildew resistance, drought and heat tolerance, photo-thermo-insensitivity, preharvest sprouting	[64,65,66]
Blackgram (*Vigna mungo* (L.) Hepper)	Hot humid, semiarid regions	Short duration, MYMV and powdery mildew resistance, photo-thermo-insensitivity, tolerance to excess moisture stress	[64,67,68]
Pigeaonpea (*Cajanus cajan* (L.) Millsp.)	Semiarid and lower humidity tropic regions	Short-to-medium duration; short stature; resistance to PSB, wilt, SMD, pod borer, and pod fly	[69,70]
Grass pea (*Lathyrus sativus* L.)	Indian subcontinent and Mediterranean region	ODAP content, water-logging and drought tolerance	[63,71]
Common bean (*Phaseolus vulgaris* L.)	Most domesticated pulse for many tropical countries	Dwarfness; resistance to CBB; tolerance to cold, heat, and drought; earliness	[63,72,73,74]
Rice bean (*Vigna umbellata* (Thunb.) Ohwi and Ohashi)	Dry zones of the arid and semiarid regions	Tolerance to acid soils and drought, early maturity, high yield, determinate growth habit	[63,75]
Tepary bean (*Phaseolus acutifolius* A. Gray)	Dry season of tropical regions	Drought and CBB resistance, deep root system, tolerant to heat, high N_2_ fixation, short growth period	[63,76,77]
Lima bean (*Phaseolus lunatus* L.)	Soils and climates of Piedmont of Georigia, Mexico, and Argentina	Plant types for marginal soil and limited water conditions, climbing types, bushy, compact types for intensive cultivation, large seed type, less cooking time	[63,78,79]
Runner bean (*Phaseolus coccineus* L.)	Cool climates of Italy and other parts	CBB resistance, high osmoregulation, heat tolerance and resistance to BCMV, dwarfness, early maturity	[63,80,81]
Adzuki bean (*Vigna angularis* Ohwi and Ohashi)	Subtropical and temperate climate zone	CBB resistance, drought tolerance	[63,82]
Hyacinth bean (*Lablab purpureus* (L.) Sweet)	Subhumid and semiarid conditions	Early maturity, drought tolerance, salinity tolerance	[63,83,84]
Horse gram (*Macrotyloma uniflorum* (Lamb.) Verds)	Low and erratic rainfall areas, better soils of the arid and semiarid regions	High tolerance towards acid soils, drought tolerance, green foliage till maturity, thermoinsensitivity, short maturity period, erect, nontendril plant type	[63,85,86]
Winged bean (*Psophocarpus tetragonolobus* (L.) D.C.)	Vietnam, parts of China	Erect type, determinate growth habit, high seed protein and oil content with high linoleic acid, photoperiodic responses	[63,87,88]
Cowpea (*Vigna unguiculata* (L.) Walp.)	Arid and semiarid regions, wide adaptation	Fast initial growth, early maturity, better source sink relations	[63,89,90]
Moth bean (Vigna aconitifolia (Jacq.) Marechal)	Arid tracts, low rainfall and warm climates	High photosynthates, tolerance to drought and heat, low fertility requirement, early and synchronous maturity, erect plant growth, tolerance to YMV	[63,91,92]

AB: *Aschochyta* blight, BGM: *Botrytis* greymold, BCMV: bean common mosaic virus, CBB: common bacterial blight, MYMV: mungbean yellow mosaic virus, ODAP: β-oxalyldiaminopropionic acid, PSB: *Phytophthora* stem blight, SMD: sterility mosaic disease, YMV: yellow mosaic virus.

**Table 2 ijms-20-02971-t002:** Genes/QTLs for adaptive traits identified in major and minor pulse crops.

Common Name	QTL/Gene	Trait	Method Used for Identification	Reference
Pea	*nod3*	Hyper nodulation mutation	Comparative genomics	[144]
	*PsMlo*	Powdery mildew resistance	Comparative genomics	[145,146]
	*PsDREB2A*	Drought response	Comparative genomics	[147]
Cowpea	Cowpea Co-like gene family	Photoperiod responsive	Sequencing along with comparative genomics	[148]
	*Stg*	Stem greenness after drought	QTL mapping	[149]
	*Rdw*	Dry weight recovery after drought	QTL mapping	[149]
	*Mac 1*–*9*	Resistance to *Macrophomina*	QTL mapping	[150]
	Major QTL	Cowpea leaf shape imparting drought tolerance	QTL mapping	[151]
	*Dro-1*, *Dro-3*, and *Dro7*	Stay-green	QTL mapping	[152]
	*Hbs-1*–*Hbs-3*	Heat-induced browning of seed coats	QTL mapping	[153]
	*Thr-1*–*Thr-3*	Foliar thrips	QTL mapping	[149]
	Major QTL	Aphid resistance	QTL mapping	[154]
	Major QTL	Resistance to root-knot nematodes	QTL mapping	[155]
	*Fot31*	*Fusarium* wilt		[151]
	Candidate genes	Resistance to root-knot nematodes	QTL mapping and transcriptome analysis	[156]
Pigeonpea	*Hsf* genes	Heat-response	Genome-wide analysis	[157]
	Dehydrin-like protein (*DLP*) gene and acid phosphatase class B family protein (*APB*) gene	Drought stress	Differentially expressed genes analysis	[158]
	Cyclophilin (*CcCYP*) gene	Multiple abiotic stress tolerance	cDNA expression analysis	[159]
	Pre-hevein-like protein PR-4 precursor (*PR-4*) and protease inhibitor/seed storage/LTP family protein (*Ltp*) genes	Defense against *Helicoverpa armigera*	Gene expression analysis using qPCR	[160]
Common bean	*Co-1*–*Co-10*	Resistance to anthracnose	Linkage mapping	[161]
	10 QTLs/genes	Resistance to anthracnose	Associations mapping	[143]
	Resistance gene analogs	Resistances to different pathogens	Associations mapping	[162]
Horse gram	9 genes	Response to drought stress	Transcriptome analysis	[163]
Adzuki bean	*VaAGL*, *VaPhyE*, and *VaAP2*	Flowering time and pod maturity	QTL mapping	[164]
Hyacinth bean	17 functionally relevant genes	Drought-stress response	Suppression subtraction hybridization (SSH) analysis	[83]
Chickpea	Aquaporins gene family	Biotic and abiotic stresses	Comprehensive genome-wide analysis	[165]
	*CarERF116*	Abiotic stress responsive	Genome-wide association analysis	[166]
	Major QTLs corresponding to flowering time genes (*efl-1*, *efl-3*, and *efl-4*)	Flowering time	QTL mapping	[167]
	*CarLEA4*	Plant developmental processes and abiotic stress responses	Gene expression analysis	[168]
	Differentially expressed genes	Drought stress response	Quantitative real-time PCR (qRT-PCR) analysis	[169]

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
