# Peer review of "Towards Exploitation of Adaptive Traits for Climate-Resilient Smart Pulses"

_ijms, 2019, doi:10.3390/ijms20122971_

Round 1

Reviewer 1 Report

Considerable efforts are required to improve the title of the manuscript. "genomic-based exploitation of adaptive" - this can be simplified. 

These are primarily cultivated on marginal lands with little inputs in several resource-poor countries of the world including South Asia. As a result, pulse crops face several biotic and abiotic stresses. - inappropriate and misleading. You can change it to " significant losses can occur due to abiotic and biotic stress. 

"However, the genetic diversity of pulse crops" - existing genetic diversity. 

Instead of enlisting authors can provide a table or refer an existing table for the names of pulse crops. 

"These are cultivated ..... These are an important source of protein .... These are consumed as grain and green vegetables." Very redundant and passive writing. Authors need to improve the readability of the article. 

The first four paragraphs of the introduction need to be condensed down to half. Make only two paragraphs. 

Presently, the article is very redundant therefore it is difficult to note down all the suggestions for each of the paragraphs. I suggest for extensive editing to make it concise. The present form is more like a book chapter. 

Author Response

Response to Reviewer 1 Comments

Point 1. English language and style: Extensive editing of English language and style required 

Response 1. Done as per suggestion

Point 2. Considerable efforts are required to improve the title of the manuscript. "genomic-based exploitation of adaptive" - this can be simplified. 

Response 2. Title has been simplified to “Towards exploitation of adaptive traits for climate resilient smart pulses

Point 3. These are primarily cultivated on marginal lands with little inputs in several resource-poor countries of the world including South Asia. As a result, pulse crops face several biotic and abiotic stresses. - inappropriate and misleading. You can change it to " significant losses can occur due to abiotic and biotic stress.

Response 3. We have made necessary changes to make it (sentence) appropriate and meaningful. 

Point 4. "However, the genetic diversity of pulse crops" - existing genetic diversity. 

Response 4. We have made necessary substitution. 

Point 5. Instead of enlisting, authors can provide a table or refer an existing table for the names of pulse crops. 

Response 5. We have done the needful as per suggestion. 

Point 6. "These are cultivated ..... These are an important source of protein .... These are consumed as grain and green vegetables." Very redundant and passive writing. Authors need to improve the readability of the article.

Response 6. Necessary changes have been incorporated to improve the readability of the article.

Point 7. The first four paragraphs of the introduction need to be condensed down to half. Make only two paragraphs. 

Response 7.  

i. Introductory portion comprises three paragraphs only. After incorporating the suggestion listed at S. No. 5, it is condensed to some extent.

ii. Because of suggestion from other reviewer to add a few sentences on adverse impact of climate change on production of pulses, and to list out biotic and abiotic stresses caused by climate change in the introductory section, it is very difficult to condense down it further.

iii. However, attempts were made to address the issue as far as possible.

Point 8. Presently, the article is very redundant therefore it is difficult to note down all the suggestions for each of the paragraphs. I suggest for extensive editing to make it concise. The present form is more like a book chapter. 

Response 8. Extensive editing has been done to make it concise.

Reviewer 2 Report

This review deals with breeding efforts and opportunities involving a range of legumes that are important protein sources and major crops for countries where food security is an increasingly pressing problem as climate instability intensifies.  In this regard, the review could be a very useful vehicle for educating the agricultural community regarding the state of legume breeding, genetics and genomics, and the opportunities for accelerating progress in crop resilience.  However, there are several weaknesses in the review as currently configured that I believe limits its usefulness.

It is not clear who the authors are targeting as an audience for this review. If it is the legume breeding community, then the review is weak in reviewing modern technologies and their applicability to legumes. For example, there is little discussion about the current status of genomics resources for legumes, the extent that Medicago and soybean resources might facilitate breeding and assessment of genetic diversity resources. In fact, there is almost nothing to educate the reader to what has been learned about the legume genome. This overview, and the state of the art in genomics of legume models, would have helped the legume breeder understand what resources are available, how to exploit them, and what is needed.

There is also relatively little discussion of breeding innovations, strategies for accelerating the breeding process, or examples of effective approaches in other systems that might be a model for bean. One approach to enhancing heterozygosity in legumes, for example, is introduction of male sterility systems, but these are not mentioned.  Many of these legumes are facultatively gynodioecious, which can be valuable for enhancing diversity if manipulated effectively in a population. There is very little detail regarding options for transgenic or cisgenic improvement. Are these legumes amenable to transformation? Are there examples of successful transgenic trait deployment in legumes? Because many of these crops are important in the developing world, will transgenic approaches be accepted? These are all details that are essential to the legume breeding audience.

The review provides a fairly superficial overview of numerous topics without meaningful detail on any one. In this way, the text appears scattered, jumping from one subject to the next without conveying any clear conclusions or definitive information. There is considerable text spent on “genotypic adaptation” which appears to be an awkward wording for what standard breeding has comprised for hundreds of years.  As a consequence, it is not clear what new information is being conveyed in this section. Do the authors mean to convey novel approaches to selection for adaptive variation, or do they have information on the underlying genetics and innovative transfer strategies for these traits?

Much of the emerging literature on environmental resilience and phenotypic plasticity involves epigenomics. It is now feasible to conduct genome-wide methylome sequencing at single nucleotide resolution. There is reason to believe that integration of DNA methylation and gene expression data may provide insight into gene networks that contribute to resiliency. Would it not be worthwhile to contemplate epigenomic variation and how to exploit this? Are there examples of chromatin immunoprecipitation and/or methylome studies that have provided insight into sources of resiliency that might be applied to legumes? Certainly methylome analysis has been conducted in soybean that might be of value to other legumes.

To simply survey RNAseq datasets is not particularly useful without the benefit of spatio-temporal and network enrichment analysis; yet the authors suggest that transcriptomic surveys have provided meaningful insight.  In fact, many of the studies cited in proteomics, transcriptomics and metabolic profiling are largely descriptive and have not demonstrated anything meaningful in terms of causality.  The review would benefit from more detailed assessment of the literature to date; citation without the benefit of insightful analysis is not very useful. There has been important progress in these areas in other species that might be valuable to legume improvement prospects.  The intense focus on pulse crops without the benefit of a few examples from other, better resourced plant models limits the scope and impact of the review.

The manuscript needs significant editing for English usage and writing style. It is poorly written and is grammatically wanting.  

Author Response

Point 1. English language and style: Extensive editing of English language and style required 

Response 1. Authors have done the needful to the best possible extent.

Point 2. This review deals with breeding efforts and opportunities involving a range of legumes that are important protein sources and major crops for countries where food security is an increasingly pressing problem as climate instability intensifies.  In this regard, the review could be a very useful vehicle for educating the agricultural community regarding the state of legume breeding, genetics and genomics, and the opportunities for accelerating progress in crop resilience.  However, there are several weaknesses in the review as currently configured that I believe limits its usefulness.

Response 2. The review deals with exploitation of adaptive traits for developing climate resilient smart pulses (grain legumes) through integration of various approaches. Authors have tried to remove a few weaknesses which other reviewers have also pointed out.

Point 3. It is not clear who the authors are targeting as an audience for this review. If it is the legume breeding community, then the review is weak in reviewing modern technologies and their applicability to legumes. For example, there is little discussion about the current status of genomics resources for legumes, the extent that Medicago and soybean resources might facilitate breeding and assessment of genetic diversity resources. In fact, there is almost nothing to educate the reader to what has been learned about the legume genome. This overview, and the state of the art in genomics of legume models, would have helped the legume breeder understand what resources are available, how to exploit them, and what is needed.

Response 3. It is very much clear that authors have targeted pulse breeding community as their target group. It was beyond the scope of this review to discuss comprehensively the genomics resources of Medicago and soybean. However, authors have discussed draft genomics resources available in chickpea, pigeonpea, mungbean, etc. and suggested the ways and mechanism to exploit these information for developing climate smart pulses. 

Point 4. There is also relatively little discussion of breeding innovations, strategies for accelerating the breeding process, or examples of effective approaches in other systems that might be a model for bean. One approach to enhancing heterozygosity in legumes, for example, is introduction of male sterility systems, but these are not mentioned.  Many of these legumes are facultatively gynodioecious, which can be valuable for enhancing diversity if manipulated effectively in a population. There is very little detail regarding options for transgenic or cisgenic improvement. Are these legumes amenable to transformation? Are there examples of successful transgenic trait deployment in legumes? Because many of these crops are important in the developing world, will transgenic approaches be accepted? These are all details that are essential to the legume breeding audience.

Response 4. It is very unfair to say that this review contains relatively little discussion of breeding innovations. Pulses are cleistogamous crops with varying degree of insects’ aided natural outcrossing, thus creating diversity in natural population. This fact is already mentioned in the introductory section. This review already comprises a separate subsection (5.1.4) on transgenics. Moreover, the review contains answers to many a question raised by the respected reviewer. 

Point 5. The review provides a fairly superficial overview of numerous topics without meaningful detail on any one. In this way, the text appears scattered, jumping from one subject to the next without conveying any clear conclusions or definitive information. There is considerable text spent on “genotypic adaptation” which appears to be an awkward wording for what standard breeding has comprised for hundreds of years.  As a consequence, it is not clear what new information is being conveyed in this section. Do the authors mean to convey novel approaches to selection for adaptive variation, or do they have information on the underlying genetics and innovative transfer strategies for these traits?

Response 5. Authors do not agree with the views expressed by the respected reviewer. We have tried our best to put the sections and subsections in a highly correlated manner to provide for conveying definitive message. As the review deals with exploitation of adaptive traits, it is quite natural to devote considerable text to “genotypic adaptation”. 

Point 6. Much of the emerging literature on environmental resilience and phenotypic plasticity involves epigenomics. It is now feasible to conduct genome-wide methylome sequencing at single nucleotide resolution. There is reason to believe that integration of DNA methylation and gene expression data may provide insight into gene networks that contribute to resiliency. Would it not be worthwhile to contemplate epigenomic variation and how to exploit this? Are there examples of chromatin immunoprecipitation and/or methylome studies that have provided insight into sources of resiliency that might be applied to legumes? Certainly methylome analysis has been conducted in soybean that might be of value to other legumes.

Response 6. In this review, there is already a subsection (5.2.2) on epigenomics for improving phenotypic plasticity to climate changes. Answers to respected reviewer’s questions are already mentioned in this subsection. However, authors have incorporated a few sentences keeping in view of his suggestion about methylome studies towards resiliency in legumes/pulses.

Point 7. To simply survey RNAseq datasets is not particularly useful without the benefit of spatio-temporal and network enrichment analysis; yet the authors suggest that transcriptomic surveys have provided meaningful insight.  In fact, many of the studies cited in proteomics, transcriptomics and metabolic profiling are largely descriptive and have not demonstrated anything meaningful in terms of causality.  The review would benefit from more detailed assessment of the literature to date; citation without the benefit of insightful analysis is not very useful. There has been important progress in these areas in other species that might be valuable to legume improvement prospects.  The intense focus on pulse crops without the benefit of a few examples from other, better resourced plant models limits the scope and impact of the review.

Response 7. The review is on pulse crops; therefore, the focus was on this group of crops. Wherever possible, other legume species including soybean was already exemplified in the text. 

Point 8. The manuscript needs significant editing for English usage and writing style. It is poorly written and is grammatically wanting.

Response 8. It is unfair to say that the review is poorly written. However, we have corrected minor mistakes in the text to the best possible extent.

Reviewer 3 Report

The review is well written.

Period needs to be added in line 39 after '[1]' and line 64 after '[9]'.

In Introduction, please add a few sentences to extensively discuss the previously reported adverse impact of climate change on production of pulse. Also, please list out the biotic stress and abiotic stress caused by climate change, instead of only emphasize on the drought stress.

In '2. Overview of Adaptive Traits in Pulses', only adaptive traits for drought stress is extensively reviewed, please also add a few sentences to cover the adaptive traits for other stress triggered by climate change.

In '5.1.2. Gene Based Markers Related to Adaptive Traits', the genes instead of the gene based marker were reviewed, either change the paragraph title or change the content to fit the title. Secondly, please list out the specific gene expressing the adaptive traits instead of only list the total number of genes.

Please combine 5.1.1. with 5.1.3. 

Author Response

Point 1. English language and style: English language and style are fine/minor spell check required.

Response 1. Authors have performed minor spell check as per suggestion.

Point 2. The review is well written.

Response 2. Authors appreciate positive comment on the MS made by the respected reviewer.

Point 3. Period needs to be added in line 39 after '[1]' and line 64 after '[9]'

Response 3. We have corrected the mistakes. 

Point 4. In Introduction, please add a few sentences to extensively discuss the previously reported adverse impact of climate change on production of pulse. Also, please list out the biotic stress and abiotic stress caused by climate change, instead of only emphasize on the drought stress.

Response 4. We have added a few sentences dealing with adverse impact of climate change on pulses. We have also listed out biotic and abiotic stresses caused by climate change in the introductory section. 

Point 5. In '2. Overview of Adaptive Traits in Pulses', only adaptive traits for drought stress is extensively reviewed, please also add a few sentences to cover the adaptive traits for other stress triggered by climate change.

Response 5. We have addressed the concerns to the best possible extent. 

Point 6. In '5.1.2. Gene Based Markers Related to Adaptive Traits', the genes instead of the gene based marker were reviewed, either change the paragraph title or change the content to fit the title. Secondly, please list out the specific gene expressing the adaptive traits instead of only list the total number of genes.

Response 6. The paragraph title has been changed as per suggestion. Specific gene expressing the adaptive traits has been listed.

Point 7. Please combine 5.1.1. with 5.1.3.

Response 7. We have combined the subsection 5.1.1 with 5.1.3.

Round 2

Reviewer 1 Report

The revised version still need significant efforts to get the article ready to publish.

As I mentioned in the first round of revision, the review is much like a book chapter which need not provide critical discussion. For review, every section needs to discuss critical issues and provide conclusive remark. Mere descriptive information will not serve the purpose of the review article. Many of the sections are more descriptive. 

Title and the objective stated in the abstract are not matching. -"This review discusses genomics‐based approaches and strategies to exploit adaptive traits for breeding climate‐smart pulses"

Table 1 and Table 2 are redundant authors can merge those and make a single table. 

Author Response

Point 1.English language and style: Moderate English changes required . 

Response 1. Done as per suggestion.

Point 2. The revised version still need significant efforts to get the article ready to publish.

Response 2. We have made significant efforts to get the article ready to publish. 

Point 3. As I mentioned in the first round of revision, the review is much like a book chapter which need not provide critical discussion. For review, every section needs to discuss critical issues and provide conclusive remark. Mere descriptive information will not serve the purpose of the review article. Many of the sections are more descriptive.

Response 3. We have made necessary changes in the second round of revision, making it look like a review, and not a book chapter. Conclusive remarks have been added in each section.

Point 4. Title and the objective stated in the abstract are not matching. -"This review discusses genomicsbased approaches and strategies to exploit adaptive traits for breeding climatesmart pulses"

Response 4. We have obviated these deficiencies from both Abstract and Introductory sections.

Point 5. Table 1 and Table 2 are redundant authors can merge those and make a single table.

Response 5. We have merged table 1 with Table 2 with desired modifications. 

Reviewer 2 Report

The review article has improved slightly by the addition of a few clarifying statements. However, the authors have elected to ignore most of my suggestions and critique, and the revised version continues to display the following weaknesses:

-The text appears to have been prepared in haste, and requires editing for English grammer, typographical errors and poor sentence structure. This problem is more pronounced in the newly edited text. 

The authors cite references with seemingly little attention to providing specific insight to the research being cited or offering vision for future prospects in legume improvement. There is little direct reference to meaningful data from the references cited in many sections, instead including only general summary statements. The newly added references on DNA methylation appear to be random, providing little information, and this is the case for most of the CRISPR references as well.  Does methylation data provide insight into gene networks that are important to stress response? Can methylation marks be targeted as markers? What would also be useful is some discussion about how CRISPR could be used directly to benefit legume breeding (what genes could/would be targeted? What is the state of legume transformation and efficacy of CRISPR in legumes?) Are there specific challenges to these technologies for crop improvement in pulses? 

  The authors appear to give equal weight to the various studies cited. They imply, in some cases, that studies that list gene expression changes in response to stress equate to stress tolerance pathways, which is misguided. Many of the references cited for stress-induced gene expression do not address networks conferring tolerance. These sections need to be more carefully worded. It would be useful for the authors to describe some of the networks that appear most prominent for abiotic and biotic stress tolerance in legumes, and whether they have shown amenability to selection.  An abundance of soybean literature in this regard has been ignored, and the authors similarly ignore work in Medicago that might have been valuable in identifying gene targets. It might be useful to explain why these omissions have been made in favor of less robust studies in the legumes chosen.

Author Response

Point 1. English language and style: Moderate English changes required.

Response 1. Authors have made the necessary changes as desired by the respected reviewer.

Point 2. The review article has improved slightly by the addition of a few clarifying statements. However, the authors have elected to ignore most of my suggestions and critique, and the revised version continues to display the following weaknesses:

Response 2. We appreciate the reviewer for his positive statement on the efforts taken by the authors to improve the article as per suggestion.

Point 3. The text appears to have been prepared in haste, and requires editing for English grammer, typographical errors and poor sentence structure. This problem is more pronounced in the newly edited text.

Response 3. Respected reviewer is right in the sense that authors were given only 10 days time for revision. We had to address the concerns of all the three reviewers. Though we took much pains to improve the English and sentence structure, we could have missed some typographical errors. However, we have tried our best to address the aforesaid shortcomings.  

Point 4. The authors cite references with seemingly little attention to providing specific insight to the research being cited or offering vision for future prospects in legume improvement. There is little direct reference to meaningful data from the references cited in many sections, instead including only general summary statements. The newly added references on DNA methylation appear to be random, providing little information, and this is the case for most of the CRISPR references as well.  Does methylation data provide insight into gene networks that are important to stress response? Can methylation marks be targeted as markers? What would also be useful is some discussion about how CRISPR could be used directly to benefit legume breeding (what genes could/would be targeted? What is the state of legume transformation and efficacy of CRISPR in legumes?) Are there specific challenges to these technologies for crop improvement in pulses? 

Response 4. We have adopted a balanced approach to address the concerns of the respected reviewer-2 in the best possible manner in the revised MS.  

Point 5. The authors appear to give equal weight to the various studies cited. They imply, in some cases, that studies that list gene expression changes in response to stress equate to stress tolerance pathways, which is misguided. Many of the references cited for stress-induced gene expression do not address networks conferring tolerance. These sections need to be more carefully worded. It would be useful for the authors to describe some of the networks that appear most prominent for abiotic and biotic stress tolerance in legumes, and whether they have shown amenability to selection.  An abundance of soybean literature in this regard has been ignored, and the authors similarly ignore work in Medicago that might have been valuable in identifying gene targets. It might be useful to explain why these omissions have been made in favor of less robust studies in the legumes chosen.

Response 5. We have given equivalent weight to various studies. Many studies performed in soybean and the model legume species have been cited, and that how these can be useful in pulses have also been discussed.  

Round 3

Reviewer 1 Report

Authors have tried to address all of my concern. 

Reviewer 2 Report

This is the third review of this manuscript. It shows reasonable improvement and may provide a useful summary of various breeding approaches used in pulses.  The manuscript will need some editing for english and spelling, but has otherwise benefitted from past revisions.